# Advanced Insights into Human Uterine Innervation: Implications for Endometriosis and Pelvic Pain

**DOI:** 10.3390/jcm13051433

**Published:** 2024-03-01

**Authors:** Audrey Astruc, Léa Roux, Fabien Robin, Ndeye Racky Sall, Ludivine Dion, Vincent Lavoué, Guillaume Legendre, Jean Leveque, Thomas Bessede, Martin Bertrand, Jules Odimba Mpoy, Emmanuel Nzau-Ngoma, Xavier Morandi, Alain Chedotal, Maela Le Lous, Krystel Nyangoh Timoh

**Affiliations:** 1Laboratoire d’Anatomie et d’Organogenèse, Faculté de Médecine, Centre Hospitalier Universitaire de Rennes, 35000 Rennes, France; audtruc@yahoo.fr (A.A.); lea.roux.internat@gmail.com (L.R.); xavier.morandi@chu-rennes.fr (X.M.); 2Department of Obstetrics and Gynecology, Rennes University Hospital, 35000 Rennes, France; nrackysall@gmail.com (N.R.S.); ludivine.dion@chu-rennes.fr (L.D.); vincent.lavoue@chu-rennes.fr (V.L.); jean.leveque@chu-rennes.fr (J.L.); Maela.LE.LOUS@chu-rennes.fr (M.L.L.); 3Department of Obstetrics and Gynecology, Angers University Hospital, 49100 Angers, France; g_legendre@hotmail.com; 4H2P2 Histopathology Laboratory, Rennes 1 University, 35000 Rennes, France; fabien.robin@chu-rennes.fr; 5Department of Hepatobiliary and Digestive Surgery, University Hospital, Rennes 1 University, 35000 Rennes, France; 6INSERM U1242, Chemistry Oncogenesis Stress Signaling, Rennes 1 University, 35000 Rennes, France; 7INSERM, LTSI—UMR 1099, Rennes 1 University, 35000 Rennes, France; 8INSERM, IRSET—UMR_S 1085, 35000 Rennes, France; 9Urology Department, APHP, Université Paris-Saclay, 94270 Le Kremlin-Bicetre, France; thomas.bessede@gmail.com; 10Surgery Department, Nîmes University Hospital, University of Montpellier, 30900 Nîmes, France; martinmbertrand@yahoo.fr; 11Department of Obstetrics and Gynecology, University Hospital of Kinshasa, Kinshasa, Democratic Republic of the Congo; odimbajules1006@gmail.com (J.O.M.); emdango@gmail.com (E.N.-N.); 12INSERM, CNRS, Institut de la Vision, Sorbonne Université, 75012 Paris, France; alain.chedotal@inserm.fr

**Keywords:** anatomy, innervation, endometriosis, pelvic pain

## Abstract

(1) **Background**: Understanding uterine innervation, an essential aspect of female reproductive biology, has often been overlooked. Nevertheless, the complex architecture of uterine innervation plays a significant role in conditions such as endometriosis. Recently, advances in histological techniques have provided unprecedented details about uterine innervation, highlighting its intricate structure, distribution, and density. The intricate nature of uterine innervation and its influence on pathologies such as endometriosis has garnered increasing attention. (2) **Objectives**: This review aims to compile, analyze, and summarize the existing research on uterine innervation, and investigate its implications for the pathogenesis of endometriosis and associated pain. (3) **Methods**: A systematic review was conducted in line with PRISMA guidelines. Using the PubMed database, we searched relevant keywords such as “uterine innervation”, “endometriosis”, and “pain association”. (4) **Results**: The initial literature search yielded a total of 3300 potential studies. Of these, 45 studies met our inclusion criteria and were included in the final review. The analyzed studies consistently demonstrated that the majority of studies focused on macroscopic dissection of uterine innervation for surgical purposes. Fewer studies focused on micro-innervation for uterine innervation. For endometriosis, few studies focused on neural pain pathways whereas many studies underlined an increase in nerve fiber density within ectopic endometrial tissue. This heightened innervation is suggested as a key contributor to the chronic pain experienced by endometriosis patients. (5) **Conclusions**: The understanding of uterine innervation, and its alterations in endometriosis, offer promising avenues for research and potential treatment.

## 1. Introduction

A deep and precise understanding of pelvic neuroanatomy, particularly concerning the uterus, is indispensable for optimal surgical practice, focusing on nerve preservation and a comprehensive grasp of pain mechanisms. Nerve injuries are closely linked to postoperative functional sequelae [1]. Furthermore, the pathophysiological processes of benign diseases like endometriosis significantly involve neuroanatomical pathways, where alterations in the neural architecture are evident [2,3]. Approximately 10% of women globally are affected by endometriosis, a condition marked by notable neuroanatomical plasticity.

Recent years have witnessed substantial progress in pelvic neuroanatomy, driven by advancements in technologies such as immunohistochemistry, fluorochromes, computer-assisted anatomical dissection (CAAD), and notably, the three-dimensional reconstruction of uterine micro-innervation [4,5]. Despite these developments, a comprehensive understanding of uterine innervation, crucial to female reproductive biology, has often been overlooked. The intricacies of intrauterine innervation’s organization and specificity, along with its interactions with the uterus’s various parts and layers, are yet to be fully deciphered.

The complex nature of uterine innervation, particularly its impact on pathologies like endometriosis, has increasingly come under scrutiny. While nerve-sparing techniques in endometriosis surgery have shown promise in mitigating postoperative functional disorders, such as voiding dysfunction, their effectiveness is not yet at an optimal level [6,7,8]. Additionally, the diverse symptomatology exhibited by patients with endometriosis might be attributable to anomalies in uterine innervation.

In light of these considerations, this study embarks on a systematic and comprehensible analysis of the existing literature on human uterine innervation. The aim is to uncover valuable insights for enhancing pelvic nerve-sparing techniques and deepening our understanding of female pelvic pain. This endeavor is not only a pursuit of academic interest but a necessary step towards improving surgical outcomes and the quality of life for women afflicted with conditions influenced by uterine innervation.

## 2. Materials and Methods

### 2.1. Search Strategy

We conducted a systematic review according to the Preferred Reporting Items for Systematic Reviews and Meta-analysis guidelines (PRISMA) [9]. Two investigators (Astruc Audrey and Roux Léa) performed an English literature search on Medline (Pubmed) from January 1946 to March 2022. We selected articles in English including uterine innervation in human bodies. The following keywords were used: uterine nerves anatomy OR uterine innervation OR uterus innervation AND anatomy and histology OR hypogastric plexus AND female OR innervation AND pelvis AND female. For the relationship with endometriosis, we used the following keywords: uterine innervation AND endometriosis AND pain association. After the exclusion of duplicate articles, all the articles were screened by the two investigators. Titles and abstracts were initially assessed for eligibility before conducting a second selection based on the full text to exclude inappropriate articles. Any discrepancies were resolved by consensus.

### 2.2. Inclusion and Exclusion Criteria

Studies were included if they described human uterine micro and macro-anatomy (adult or fetal). We also included articles describing the association between uterine innervation anomaly and endometriosis, or pelvic pain.

We excluded commentaries, editorials, expert consensus, reviews, abstracts.

### 2.3. Data Extraction (Figure 1)

The principal data items extracted and analyzed from the articles were as follows:-Macroscopic anatomical description;-Microscopic anatomical description;-Surgical description;-Population: adults, fetuses.

**Figure 1 jcm-13-01433-f001:**
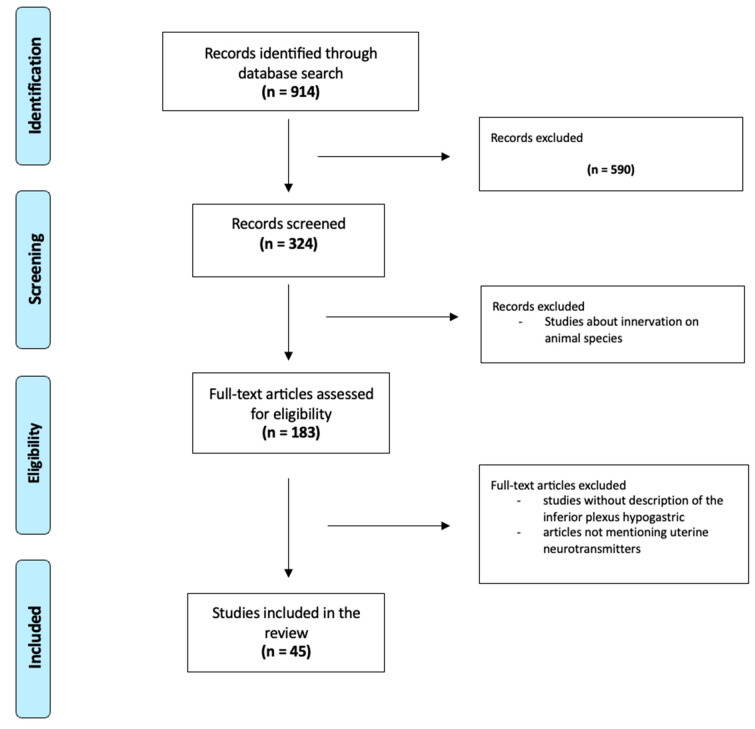
Flow Chart.

## 3. Results

### 3.1. Uterine Macro Innervation

The purpose of this section is to provide a comprehensive and detailed description of the Inferior Hypogastric Plexus (IHP), including its inputs, outputs, location, shape, and functional roles. As an intermediate nervous structure between the central nervous system and the visceral organs, it is critical to understand that it has descending visceromotor inputs and outputs and ascending viscerosensitive inputs and outputs, regardless of the direction of the description. We will use the usual craniocaudal description in which nerves above the IHP are often referred to as afferent to the IHP (roots), and nerves below the IHP as efferent from the IHP (branches). This conception is only valid for the descending visceromotor fibers. It must be reversed regarding the ascending viscerosensitive fibers.

#### 3.1.1. Macro Anatomical Distribution

(a)Nerves above the IHP (roots)

The IHP, also known as the pelvic plexus, is an intricate network formed by the convergence of several nerves: the hypogastric nerves, which arise from the superior hypogastric plexus; the pelvic splanchnic nerves, which arise from the anterior branches of the sacral roots; and the sacral splanchnic nerves, which arise from the sacral sympathetic trunks.

Regarding the pelvic splanchnic nerves: the S1 root is not involved according to some authors [7,10], while the S4 root seems to be the most important [10].

Regarding the sacral splanchnic nerves: they come from the S2 to S3 sympathetic ganglia, pass through the utero-sacral ligaments and join the IHP, forming the dorsal edge of the IHP [5,7].

The bilateral hypogastric nerves follow a distinct course, passing through the presacral space and ascending along the rectum to reach the mesorectal fascia, navigating via the lateral rectal ligament [10]. These nerves run dorsally along the ureter, creating a specific anatomical relationship that is critical for surgical considerations [5,7].

Notably, variations in the trajectory and positioning of these nerves have been observed and documented. Seracchioli et al. noted a differential positioning of the right hypogastric nerve compared to the left, with the right being situated further from the ureter but closer to the mid-cervical plane and the utero-sacral ligament [11]. Fermaut et al. identified the hypogastric nerve intersecting the utero-sacral ligament, which resides within the pelvic visceral fascia [7]. In contrast, Kavallaris et al. observed the hypogastric nerve running parallel to the utero-sacral ligament [12].

Ercoli et al. provided a detailed description of the hypogastric nerves, noting that, in the majority of cases (73%), these nerves present as a single bundle of fibers located medially and inferiorly to the ureter, approximately 10 mm away [6]. In 17% of cases, these nerves are observed as multiple nerve fibers oriented towards the ureter. Adding to the complexity, Aurore et al. described an additional medial accessory hypogastric nerve, which presents challenges in characterization due to its variable presence [10].

Unique among pelvic plexuses, the IHP is characterized by its bilateral symmetry, a feature that distinguishes it as the sole bilateral plexus in the human body. This bilateral aspect is crucial for its functionality, ensuring a coordinated and balanced innervation to the structures of the pelvis and perineum [13].

The IHP has a complex and variable anatomical structure. Typically, it is characterized as a fan-shaped quadrangular formation, bordered by upper, lower, anterior, and posterior edges, and defined by four distinct angles: superior–anterior, infero-anterior, postero-superior, and postero-inferior [5]. Alternatively, some authors describe it as assuming a triangular configuration, with its base oriented posteriorly and infero-anteriorly [14]. This diversity in shape reflects the intricate nature of the IHP.

(b)Location of the IHP

The IHP has a remarkably consistent anatomical positioning within the pelvic region. It is strategically situated lateral to the rectum and vagina, nestled beneath the point where the uterine artery intersects with the ureter in the pararectal fossa [5,7,8,10]. This specific location places it within the paracervix, an area critical for understanding various pelvic pathologies and surgical procedures.

The point of intersection, where the uterine artery crosses the ureter, defined as the parametrium [15], forms a significant anatomical feature, creating what is referred to as the roof or the anteroinferior angle of the IHP [14,16]. The IHP is typically found positioned under the deep uterine vein, further emphasizing its deep-seated placement within the pelvic anatomy [10,17]. Following its path, the IHP extends along the lateral wall of the rectum, then courses through the cervix and the fornix vaginalis, which corresponds to the uppermost part of the vagina, and ultimately reaches the lateral aspect of the vagina and bladder. The density of the IHP fibers varies, with the highest concentration observed either in the vaginal fornix or the paracervix, depending on different anatomical studies and interpretations [5,17]. Some authors report that, from the origin of the utero sacral ligament and the cardinal ligament in the pelvic side wall to their insertion in the uterus, the total nerve density decreases [18,19,20]. The nerves fibers are more abundant in the lateral third from the uterine cervix, in the deepest part [18]. In the vesico uterine ligament, the sympathetic and parasympathetic nerves are located in the medial side of the vesical veins. We also observed that the human uterine artery is also richly innervated [21,22,23,24].

In its unique position within the pelvic landscape, the IHP is frequently intersected by the middle rectal artery, just before its merges into the rectal wall. This juncture, along with the crossing points of the internal iliac artery branches over the plexus, serves as a crucial anatomical landmark, essential for surgical navigation and understanding pelvic neuroanatomy [5,6,10].

(c)Nerves below the IHP (branches)

The efferent pathways of the IHP constitute a complex network, essential for the innervation of various pelvic organs [5,10]. These pathways include an anterolateral branch, which serves multiple crucial functions. This branch innervates the uterus and lower bladder, dividing into a cranial segment that runs along the uterine artery through the parametrium for uterine innervation, an anterior segment that follows the path of the ureter and vagina to reach the lower bladder, and a lower segment extending to the lower rectum and *perineal* erectile bodies. This intricate branching pattern ensures thorough coverage, crossing the parametrium and vesico-uterine ligament and extensively encompassing the paracervix and its adjacent paravaginal area. In addition to the anterolateral branch, there is a posteromedial branch that targets the posterolateral aspect of the rectum after passing through the mesorectum and partially innervates the smooth anal sphincter. This branch gives rise to the inferior rectal plexus, which splits into two major perineal projections: one anteriorly towards the urethral sphincter complex and another inferiorly towards the perineal erectile bodies.

The efferent neural framework of the IHP is further delineated by various authors, who describe three distinct efferent plexuses: the rectal–vaginal plexus focusing on the vaginal and rectal regions, the vesical plexus dedicated to bladder innervation, and the inferior rectal plexus, which supplies the urethral sphincter and perineal erectile bodies. Moreover, as highlighted by Moszkowicz et al., the uterovaginal plexus emerges as a key player, innervating the erectile tissues of the vestibular bulbs and clitoris, as well as the superior vestibular glands, while the vesico-ureteral plexus and the middle rectal plexus contribute significantly to the overall functionality and sensory network of the pelvic area [5].

#### 3.1.2. Functional Distribution

It is important to recognize that the autonomic nervous system operates as a branch in parallel with the somatic system. This branching is exemplified in the structure and function of the IHP. The IHP receives its sympathetic innervation primarily from the sacral ganglia S2 to S4 [7,10], while its parasympathetic innervation is derived from the pelvic splanchnic nerves [7]. Notably, the hypogastric nerves, which form an integral part of the IHP, are unique in that they carry both sympathetic and parasympathetic fibers, receiving afferent inputs from the superior hypogastric plexus [20]. This dual sympathetic and parasympathetic nature is a characteristic feature of the IHP, as it is for all autonomic plexuses [20]. In terms of pain transmission, the pathway predominantly follows the parasympathetic route via the pelvic splanchnic nerves [25], highlighting the complex interplay between these two components of the autonomic system in the modulation of pelvic pain.

In the human uterine artery, sympathetic nerve fibers are predominantly located at the media–adventitial border of the arterial wall [23,25], playing a key role in vascular regulation. However, the presence of sensory innervation in this region remains controversial and not clearly established [22,26].

Regarding the uterine ligaments, the cardinal ligaments primarily receive parasympathetic innervation at their pelvic wall insertion points, with some sympathetic innervation in their middle and lower parts [18,19,20]. The utero-sacral ligaments on the other hand, are mostly innervated by sympathetic nerves, supplemented by a few parasympathetic and sensory fibers [18,19,27], indicating their importance in pelvic functions.

#### 3.1.3. Functional Implications

Branches of the IHP play a critical role in the functioning of the pelvic and perineal regions, so much so that the IHP can be aptly described as the “commander in chief” of these areas. However, this pivotal role also implies that any damage to the IHP, particularly during surgical procedures, can lead to significant postoperative complications.

One key aspect of understanding the IHP’s role is recognizing the potential symptoms that may arise following its lesion during surgery. High-risk areas for nerve damage include the retro-rectal space, the utero-sacral ligaments where the uterovaginal plexus traverses, as well as the vesico-uterine ligaments and the paracervix [10]. Additionally, the region where the ureter intersects with the uterine artery marks a critical boundary—the upper limit of the IHP and the commencement of its efferent pathways. Surgical dissection in and around this area carries an elevated risk of damaging the IHP [14,16].

The spectrum of symptoms associated with IHP damage is diverse, often including functional impairments such as a lack of relaxation in the detrusor and urethral sphincter muscles, leading to urinary difficulties. Other potential issues include reduced vaginal lubrication [7,25]. These symptoms highlight the IHP’s role in regulating various autonomic functions in the pelvic area.

The sympathetic fibers of the IHP, for instance, have a vasomotor function, control smooth muscle sphincters, and inhibit peristalsis. A study by Kavallaris et al. [12] underscored the consequences of not preserving these nerve fibers during surgery. Patients who underwent total hysterectomy without nerve-sparing techniques had a pronounced need for an indwelling urinary catheter and faced challenges in spontaneous micturition recovery.

### 3.2. Uterine Micro Innervation

This section is dedicated to providing a comprehensive and detailed description of the innervation within the uterus throughout its different parts and layers.

#### 3.2.1. Anatomical Distribution

The micro innervation of the uterus presents a heterogeneous and intricately patterned network. Our study focusing on human fetuses has shed light on this complexity, revealing that uterine innervation adopts a centripetal distribution, encompassing all regions and layers of the uterus—the serosa, myometrium, and endometrium [4].

Among the various parts of the uterus, the cervix stands out as the most densely innervated area [26]. This dense innervation plays a crucial role in the cervix’s functional and physiological responses, making it a key area of focus in uterine neuroanatomy.

The distribution of nerve fibers within the myometrium is particularly noteworthy. Two distinct plexuses predominantly facilitate this innervation: a subserosal plexus and another located at the pivotal endometrial–myometrial junction [28,29]. The concentration of nerve fibers is especially high at the utero tubal junction, a critical juncture for reproductive functions.

These nerve fibers are strategically positioned around neurovascular bundles, integrating seamlessly with the vascular architecture of the uterus. Within the myometrium, the innervation is moderate across both the circular and longitudinal smooth muscle layers. However, it becomes more pronounced and abundant within the vascular zone, highlighting the synergy between neural and vascular elements in uterine health and function [17,25].

In the endometrium, nerve fibers predominantly surround blood vessels at the interface with the myometrium [30]. This arrangement suggests a vital role in endometrial vascular regulation and possibly in the processes related to menstrual cycle and implantation.

#### 3.2.2. Functional Nerves Distribution

Uterine innervation encompasses both autonomic (sympathetic and parasympathetic) and sensory fibers, contributing to a multifaceted neural control system. Sympathetic innervation is mediated by neuropeptide Y (NPY) and tyrosine hydroxylase (TH) neurotransmitters, while parasympathetic innervation includes vasoactive intestinal peptide (VIP), neuronal nitric oxide synthase (NOS), and vesicular acetylcholine transporter (VaChT) neurotransmitters. Sensory innervation involves calcitonin gene-related peptide (CGRP) and substance P (SP), crucial for sensory signaling (refs).

The distribution of these nerve fibers is not uniform throughout the uterus. Sympathetic innervation is predominant in the uterine fundus [23,31,32]. The uterine body shows fewer parasympathetic fibers [30,33], with occasional instances of CGRP fibers [34,35].

In contrast, dense parasympathetic innervation is observed at the utero tubal junction and the isthmus of the uterine tube, within the smooth muscle layers, suggesting a sphincter-like function [34,36,37]. Furthermore, CGRP is present in all muscle layers of the uterine tube [38].

Sensory innervation is predominantly directed to the epithelium of the cervical region. The endometrium, however, is innervated but has a relatively sparse nerve supply [32]. The concentration of NPY is lowest here [23], with occasional sympathetic fibers being detected in the basal region between the glands and the underlying myometrium [31], along with a few CGRP fibers [38].

#### 3.2.3. Potential Functional Implications

The role of Vasoactive Intestinal Peptide (VIP) fibers in uterine physiology is diverse and significant. These fibers participate in the neural regulation of blood flow through their vasodilatory effects, influencing smooth muscle activity by inducing relaxation, and modulating secretion processes within the uterus [35,36,37].

Conversely, the adrenergic innervation in the uterus, primarily functioning as a vasoconstrictor, plays a central role in the regulation of blood flow within the human uterine artery [24]. This aspect of adrenergic activity underscores its importance in the overall vascular dynamics of the uterus.

Calcitonin gene-related peptide (CGRP), another critical component of the uterine neural network, appears to act as an inhibitor of spontaneous contractile activity in the uterus [38]. This inhibitory role of CGRP adds a layer of complexity to the intricate balance of uterine muscle dynamics and may influence various pains ranging from menstrual cramps to childbirth.

Interestingly, the distribution of these nerve fibers in the uterus is not constant throughout life. It has been observed that the density of nerve fibers tends to decrease with age. This decrease in nerve density may have implications for age-related uterine changes or functions, such as puberty, reproductive capacity, or menopausal symptoms.

#### 3.2.4. Application to Endometriosis

In examining the complexities surrounding endometriosis and adenomyosis, it becomes evident that, despite numerous studies, the underlying mechanisms of pain generation and disease progression remain only partially understood. The relationship between these conditions and uterine innervation, particularly in the context of pain, presents a field of ongoing research and debate.

Initially, studies have shown that endometriosis is linked to an increase in various types of nerve fibers, including small diameter unmyelinated and myelinated fibers (mainly sensory C fibers, as well as a mixture of sensory Adelta, adrenergic, and cholinergic fibers) in both the superficial and deep functional layers, as well as the basal layer of the endometrium [39,40,41,42]. Notably, in women with endometriosis, there appears to be an elevated presence of neuropeptide Y (NPY) and vasoactive intestinal peptide (VIP) fibers within the functional layer of the endometrium [43,44].

However, more recent research has brought new insights, revealing that women with adenomyosis or uterine fibroids, with painful symptoms, also have PGP 9.5 (Protein Gene Product) nerve fibers (a pan-neuronal marker) in the endometrium’s functional layer and myometrium. Intriguingly, these nerve fibers are absent in cases of painless adenomyosis, uterine fibroids, or endometriosis. Furthermore, no significant differences in PGP 9.5 nerve fiber density were observed between these conditions when pain was a symptom [42,45,46]. This suggests that the presence of these nerve fibers in the endometrial functional layer may be more closely associated with pain rather than the specific pathology itself.

Interestingly, the density of nerve fibers in the myometrium of women with endometriosis is found to be lower than in the endometrium but significantly higher than in women without the condition, particularly in the lower half of the uterus [40,41,43,44,47,48]. Observations have noted thick trunks of nerve fibers along the endometrial-myometrial interface or basal layer, indicating a heightened innervation. This increase in sensory C and Adelta nerves (such as SP and CGRP) suggests their role in pain generation, possibly through enhancing myometrial contractility [41,48].

In cases of deep infiltrating endometriosis (DIE), a notably rich innervation is observed, encompassing sensory Adelta, sensory C, cholinergic, and adrenergic nerve fibers (SP, VIP, TH, NPY) [49,50]. Some researchers have linked the density of these nerves to the symptoms experienced by patients [49,51].

Various theories have been proposed to explain the development of endometriosis. One such theory by Quinn et al. [52] suggests that denervation–reinnervation of the uterine fundus following nerve injury (such as postpartum or due to constipation) might be a key factor in symptom development and disease onset. Another perspective posits that, similar to malignant tumors, sensory, sympathetic, and parasympathetic nerve fibers are involved in a bidirectional interaction with endometriotic lesions. This interaction, which includes promoting inflammation, angiogenesis, proliferation, and further innervation, could contribute to the development of endometriosis [42]. Bourlev et al. have highlighted the role of the VIP neurotransmitter, particularly due to its high concentration in the endometrium and involvement in inflammation, in the generation of pain associated with endometriosis [53]. Additionally, Tokushige et al. [44,54] have shown that nerve proliferation in endometriosis could be linked to hormonal levels of estrogen and progesterone, as evidenced by a reduced nerve fiber density in the endometrium’s functional and basal layers in women with endometriosis undergoing hormonal treatment.

In contrast to endometriosis, adenomyosis is often characterized by fewer or even an absence of nerve fibers [28,29,42,48]. Notably, nerve fibers are rarely found around nerve bundles in the myometrium and there are few at the endometrial–myometrial interface. This lack of nerves in these areas may potentially contribute to the proliferation of the endometrium into the myometrium.

In summary, the role of uterine innervation in conditions like endometriosis and adenomyosis presents a complex interplay of factors. While advancements have been made in understanding the association between nerve fiber presence and pain, as well as the distribution of various types of nerve fibers in different layers of the uterus, the full extent of their roles in the pathophysiology of these conditions warrants further investigation.

## 4. Discussion

### 4.1. Synthesis of Key Findings

This review has unveiled the nuanced and complex architecture of uterine innervation, casting light on its pivotal role in gynecological conditions, notably endometriosis. The intricate distribution and density of nerve fibers in uterine tissues reveal a far more elaborate network than previously comprehended. These insights hold profound implications for clinical practice, particularly revolutionizing the surgical management of endometriosis and enhancing our understanding of pelvic pain mechanisms.

### 4.2. Strengths

The principal strength of our review lies in its exhaustive and multidimensional approach, combining advanced histological data and clinical perspectives. This approach has offered a more comprehensive understanding of uterine innervation and its role in pathologies affecting women, notably endometriosis. By integrating the latest advancements in histological techniques and clinical insights, our review provides a nuanced understanding of the complex relationship between uterine innervation and endometriosis. This is crucial for developing more effective treatments for endometriosis and other pelvic pathologies.

### 4.3. Association between Endometriosis Pain and Innervation

A significant link between increased nerve fiber density in endometriotic tissues and chronic pain experienced by patients has been identified. This correlation suggests that heightened innervation, likely influenced by inflammatory factors and neuroangiogenesis, critically contributes to the pain symptomatology in endometriosis. Understanding this link is crucial for developing targeted treatments that can more effectively manage pain in endometriosis patients. This insight also opens up new avenues for research into the mechanisms of pain generation in endometriosis, which could lead to novel therapeutic approaches.

### 4.4. Central Sensitization

When it comes to endometriosis-associated pain, it has been shown that for some patients, although the pain symptoms were relieved, these came back a few months later; this is due to central sensitization [55,56,57]. Indeed, repeated nociceptive stimuli and inflammation lead to peripherical sensitization by lowering the activation threshold of nociceptive receptors, amplifying inflammation with the secretion of substance P and CGRP [55]. The loss of sympathetic innervation is also responsible for amplifying inflammation [56]. This peripherical sensitization generates a large amount of pain stimuli at the dorsal horn of the spinal cord, remodeling it, which eventually creates central sensitization. Central sensitization is responsible for amplification and perpetuation in time of pain. We understand the urge for a better understanding of the pain generation mechanisms in endometriosis and the establishment of the right treatment early in the disease’s history. Understanding neuroanatomy is key for this.

### 4.5. Using Robotic Surgery for Nerve Preservation

A better understanding of the anatomy of women’s pelvic innervation would also help preserve it during surgery.

Nerve preservation comes into its own in surgery for deep pelvic endometriosis [7,15]. Urinary disorders are the main side effects associated with nerve tissue damage during dissection, notably via damage to the nerves of the IHP [7]. These side effects may be present or disappear several years after the operation. Taking into account the balance of benefits and risks associated with these surgeries, a genuine mapping of women’s pelvic innervation, using a standardized vocabulary applicable to all, would be a real help in taking these structures into account. Additionally, the enhanced 2D magnification provided by robotic systems could be useful in preserving nerves vital for maintaining pelvic functionality.

### 4.6. Challenge of Uterine Innervation

One of the major challenges highlighted in our review is the detailed understanding of human-specific uterine innervation and its clinical application. The individual variability and anatomical complexity pose significant challenges in symptom interpretation and surgical decision-making. Additionally, the dynamic nature of uterine innervation, influenced by hormonal changes and reproductive status, adds another layer of complexity. Understanding these nuances is essential for developing personalized treatment approaches for conditions like endometriosis and for improving surgical outcomes in nerve-sparing procedures.

### 4.7. Weaknesses of the Review

Our study is limited by the lack of direct experimental and longitudinal data, which could provide further confirmation of our observations. Additionally, variability in study methodologies and interpretation of histological data could influence the outcomes. Another limitation is the focus on published literature, which may not capture the full scope of ongoing research in this rapidly evolving field. These limitations highlight the need for continued research and collaboration across disciplines to fully unravel the complexities of uterine innervation.

### 4.8. Future Studies

Future studies should prioritize longitudinal experimental designs to more definitively establish the causal relationships between uterine innervation and endometriosis. Investigating novel treatment approaches that specifically target uterine nerve pathways holds the potential to significantly advance the management of endometriosis and its associated pain. Unanswered questions remain, particularly regarding the specific types of nerves involved, the presence of identifiable patterns, and their correlation with various symptoms. Interdisciplinary collaboration among histologists, neurologists, and gynecologists is essential for deepening our understanding and developing groundbreaking therapeutic methods.

## 5. Conclusions

This review has shed light on the intricate relationship between uterine innervation and gynecological conditions, particularly endometriosis. It highlights the critical role of uterine nerve fibers in pain perception and the potential for improved surgical outcomes with advancements like robotic system’s magnified visualization. Key challenges remain, including understanding the specific nerve types involved and their connection to various symptoms. Future research, necessitating collaboration across disciplines like histology, neurology, and gynecology, is essential to deepen our understanding and enhance treatment strategies for conditions related to uterine innervation.

## Data Availability

Not applicable.

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
