# Peer review of "Advanced Insights into Human Uterine Innervation: Implications for Endometriosis and Pelvic Pain"

_jcm, 2024, doi:10.3390/jcm13051433_

Round 1
Reviewer 1 Report
Comments and Suggestions for Authors
it is a very interesting review. please find my suggestions for minor edits below:
reference editing for line 60 and 291
unhighlight line 347 word interestingly
please add that you excluded animal studies in the materials and methods section
in your research, have you not come through any studies on biochemical markers? If you have excluded them please include in the materials and methods section and also your prisma diagram
there is an extra reference number 20 in the reference list, please edit
please inform why you haven't included studies after march 2022, since it is almost 2 years ago. please include recent research if possible.
have you registered your review to Prospero? Please provide your info
Author Response
Response to the Reviewers:
First of all, we would like to thank the Editor in Chief and the Reviewers for their comments contributing to improving our manuscript. We have answered all the reviewers’ comments point-by-point and hope that the revised manuscript is now acceptable for publication in JCM.
To Reviewer n1:
- “in your research, have you not come through any studies on biochemical markers? If you have excluded them please include in the materials and methods section and also your prisma diagram” Thank you for your remark, we indeed did not include studies on biochemical markers and have therefore corrected it in the Material and methods
- “please inform why you haven't included studies after march 2022, since it is almost 2 years ago. please include recent research if possible.” We added more recent articles to the review
- “have you registered your review to Prospero? Please provide your info” No we have not.
To Reviewer n2:
- «The conclusion are too general and do not add much. What the authors conducted it is quite widely recognized knowledge. It should be rethought and changed. » Thank you for your remark, we have indeed changed the conclusion.
Reviewer 2 Report
Comments and Suggestions for Authors
First of all, thank you for requesting to provide a review of this article, which topic is insights into human uterine innervation - Implications for Endometriosis and Pelvic Pain
There is a lot unnecessary empty pages and space, as well.
Page 2 is empty. Move introduction up.
Page 3, line 52 - clear space
Page 4. Figure 1 should be under the text.
Page 5, line 109. Write Results instead of Results (3291)
Page 7, line 212. Write distribution instead of Distribution
Page 9, line 291. Add missing reference
Page 10, line 347. Write interestingly instead of Interestingly.
Page 10, line 349. Write 40-44, 47,48 instead of 40, 41, 43, 44, 47, 48
Page 14. Half page is empty. Move next part up.
Except this linguistic errors, reading results about uterine macro and micro innervation (anatomical and functional distribution and functional implication), I have an impression that this part is too detailed with lot of already known knowledge and some parts are duplicated, as well, while part about relationship between neuroanatomy and endometriosis is too short. The number of selected articles in this review is huge enough, while information about link between neuroanatomy, endometriosis and pelvic pain are scarce. Please add more information from the selected articles, their observations and results about this, which is topic of interest of this review.
The conclusion are too general and do not add much. What the authors conducted it is quite widely recognized knowledge. It should be rethought and changed.
This article is written as review, and should be be accepted for publication after this corrections.
Author Response

(The authors gave the same response as above.)
